# Immediate postnatal prediction of death or bronchopulmonary dysplasia among very preterm and very low birth weight infants based on gradient boosting decision trees algorithm: A nationwide database study in Japan

Kota Yoneda[1], Tomohisa Seki[2], Yoshimasa Kawazoe[2,3]*, Kazuhiko Ohe[2,4], Naoto Takahashi[1], on behalf of the Neonatal Research Network of Japan[¶]

1 Department of Pediatrics, The University of Tokyo Hospital, Tokyo, Japan, 2 Department of Healthcare Information Management, The University of Tokyo Hospital, Tokyo, Japan, 3 Artificial Intelligence and Digital Twin in Healthcare, Graduate School of Medicine, The University of Tokyo, Tokyo, Japan, 4 Department of Biomedical Informatics, Graduate School of Medicine, The University of Tokyo, Tokyo, Japan

¶ Membership of the Neonatal Research Network of Japan is provided in the Acknowledgments.
* kawazoe@m.u-tokyo.ac.jp

## Abstract

### Introduction

Bronchopulmonary dysplasia (BPD) poses a substantial global health burden. Individualized treatment strategies based on early prediction of the development of BPD can mitigate preterm birth complications; however, previously suggested predictive models lack early postnatal applicability. We aimed to develop predictive models for BPD and mortality based on immediate postnatal clinical data.

### Methods

Clinical information on very preterm and very low birth weight infants born between 2008 and 2018 was extracted from a nationwide Japanese database. The gradient boosting decision trees (GBDT) algorithm was adopted to predict BPD and mortality, using predictors within the first 6 h postpartum. We assessed the temporal validity and evaluated model adequacy using Shapley additive explanations (SHAP) values.

### Results

We developed three predictive models using data from 39,488, 39,096, and 40,291 infants to predict "death or BPD," "death or severe BPD," and "death before discharge," respectively. These well-calibrated models achieved areas under the receiver operating characteristic curve of 0.828 (95% CI: 0.828–0.828), 0.873 (0.873–0.873), and 0.887 (0.887–0.888), respectively, outperforming the multivariable logistic regression models. SHAP value analysis identified predictors of BPD, including gestational age, size at birth, male sex, and

**Data Availability Statement:** Data cannot be disclosed to the public due to a license agreement and ethical issues in each participating facility of the Neonatal Research Network of Japan. For data sharing of the dataset generated during this study, please contact the corresponding author or the Artificial Intelligence and Digital Twin in Healthcare, Graduate School of Medicine, The University of Tokyo, Japan (Email: aih-office@m.u-tokyo.ac.jp), or the Office for Human Research Studies, Graduate School of Medicine and Faculty of Medicine, The University of Tokyo, Faculty of Medicine Bldg. 2 4F, 7-3-1 Hongo, Bunkyo-ku, Tokyo 113-0033, Japan (Email: ethics@m.u-tokyo.ac.jp).

**Funding:** The author(s) received no specific funding for this work.

**Competing interests:** Yoshimasa Kawazoe belongs to the "Artificial Intelligence and Digital Twin Development in Healthcare, Graduate School of Medicine, The University of Tokyo" which is an endowment department. However, the sponsors had no influence over the interpretation, writing, or publication of this work. This does not alter our adherence to PLOS ONE policies on sharing data and materials.

persistent pulmonary hypertension. In SHAP value-based case clustering, the "death or BPD" prediction model stratified infants by gestational age and persistent pulmonary hypertension, whereas the other models for "death or severe BPD" and "death before discharge" commonly formed clusters of low mortality, extreme prematurity, low Apgar scores, and persistent pulmonary hypertension of the newborn.

## Conclusions

GBDT models for predicting BPD and mortality, designed for use within 6 h postpartum, demonstrated superior prognostic performance. SHAP value-based clustering, a data-driven approach, formed clusters of clinical relevance. These findings suggest the efficacy of a GBDT algorithm for the early postnatal prediction of BPD.

## Introduction

Complications from preterm births present a significant global health challenge [1]. In 2009, it was estimated that 0.94 million of 5.3 million deaths in children under the age of five worldwide were attributed to preterm birth complications, making it the leading cause of death in this age group [2]. A study on very preterm (VPT, gestational age < 32 weeks) and very low birth weight (VLBW, birth weight < 1500 g) infants in 11 high-income countries found mortality rates of 9.1% (4.2%–15.2%) and bronchopulmonary dysplasia (BPD) incidence rates of 25.5% (10.8%–37.1%) [3]. The annual societal cost of preterm births in the United States was estimated to be $26.2 billion, or an average of $51,600 per preterm birth [4]. An additional cost of $31,565 is associated with BPD [5]. Rehospitalization and outpatient visits continue to place economic and emotional burdens on families of preterm infants [6].

Early postnatal prediction of preterm birth complications would enable the reduction of complications through alterations in treatment strategies given the evidence that advanced ventilators offering volume-targeted ventilation and/or non-invasive respiratory support can lower the risk of BPD [7, 8]. The implementation of early risk assessments would not only allow for individualized treatments but also improve the feasibility of conducting clinical trials stratified by severity and enhance the quality of parental counseling at admission. However, previously reported predictive models pose challenges in performing early postnatal predictions of BPD due to multiple reasons. First, these models include factors that are not available in the early hours after birth, such as urine output and duration of mechanical ventilation, which hinder their ability to make predictions immediately after birth. This is particularly problematic as preterm infants are most vulnerable during the first 24 h postpartum, necessitating rapid planning for resource allocation [9–19]. Second, most previous studies are based on the strong assumption that there is a linear relationship between explanatory variables and outcomes without validation. Lastly, these studies often have less than 1,000 cases and have not been validated with a sufficient number of test cases. The rarity of VLBW infants, who account for less than 1% of all live births, underlies these issues [20].

To assess the applicability of machine learning techniques in the early prediction of BPD among VPT and VLBW infants, we utilized clinical data from a domestic case registry to develop a predictive model operational within 6 h after birth. Furthermore, we examined the temporal validity of our model and assessed its adequacy using a machine learning expandability approach, focusing on the influence of individual predictors on the model's predictions.

## Materials and methods

### Data source

Data on infants with concurrent VPT and VLBW born between 2008 and 2018 were extracted from the Neonatal Research Network of Japan (NRNJ) database on June 18, 2021. Infants admitted after the first 48 h of life, those transferred during the acute phase, and those with uncertain outcomes were excluded. We did not have access to any information that could identify individual participants during or after data collection. To verify the temporal validity of the models, we divided the dataset from 2008 to 2015 into the "training and validation" set, reserving the dataset from 2016 to 2018 for the test set. The NRNJ database is a nationwide registry that collates clinical data of VPT or VLBW infants. As of 2021, it comprised 167 clinical attributes, including 29 maternal, 123 neonatal, and 15 facility-related factors. Since its establishment in 2003, the NRNJ database has registered approximately 5,000 cases of VPT or VLBW infants annually from over 200 perinatal facilities, covering over half of the perinatal facilities across Japan [21, 22].

### Outcomes and predictors

The severe cases, which have a higher risk of BPD, also exhibit higher mortality rates, leading to a "missing not at random" situation. As a practical solution, we used a composite outcome of either pre-discharge mortality or the development of BPD ("death or BPD") as our primary outcome. The definition of BPD was based on the need for respiratory support at 36 weeks postmenstrual age. We also created a prediction model for "death or severe BPD", where "severe BPD" was defined as the need for mechanical ventilation at 36 weeks postmenstrual age. To examine the contribution of pre-discharge mortality, we also developed a predictive model for mortality before discharge ("death before discharge"). From the continuously registered variables during the study period, we identified 46 predictors that were accessible within 6 h after birth, including 15 maternal, 16 neonatal, and 15 facility-related factors (**Table 1**). Respiratory distress syndrome (RDS) was diagnosed based on clinical symptoms, chest X-ray, and microbubble test. Persistent pulmonary hypertension of the newborn (PPHN) was defined as pulmonary hypertension and right-to-left shunt blood flow diagnosed by echocardiography in the acute phase. Variables with a missing rate of 30% were excluded.

### Handling of missing values

The variables incorporated in this study were essential and readily obtainable, suggesting that missing values were not closely related to the patient's conditions. Consequently, instead of categorizing missing values as a single category, we applied the multiple imputation by chained equations algorithm, using the mice package (version 3.16.0) of R (version 4.3.1) [23, 24] (**S1 Fig**). We independently created the machine-learning models with the 20 imputed "training and validation" sets and integrated these models as a single model. Similarly, we created the 20 imputed test sets and applied the machine-learning model to obtain predictions.

### The GBDT algorithm

The gradient boosting decision trees (GBDT) algorithm is a machine learning technique that works with sequential decision tree ensembles. The GBDT algorithm, with its robust expressive capabilities, provides excellent predictive performance on tabular data [25]. With its flowchart-like architecture, each tree in GBDT models intrinsically captures non-linear relationships of the outcome and variables, as well as the interactive relationships among variables. and does not require independence between variables [26]. The GBDT algorithm is a

**Table 1. Predictors and auxiliary variables.**

| Group | Variables (Data Type) |
|---|---|
| **Predictors available within 6 h of birth (46 variables)** | |
| **Maternal factor** | 1. Maternal age (Integer), 2. Gravidity (Integer), 3. Parity (Integer), 4. Number of fetuses (Integer), 5. Birth order (Integer), 6. Monochorionic multiple (Boolean), 7. Maternal diabetes (Boolean), 8. Hypertensive disorders of pregnancy (Boolean), 9. Clinical chorioamnionitis (Boolean), 10. Premature rupture of membranes (Boolean), 11. Antenatal steroid administration (Boolean), 12. Non-reassuring fetal status (Boolean), 13. Cephalic presentation (Boolean), 14. Delivery mode (Categorical: Caesarean section, Natural vaginal delivery, Forceps or vacuum delivery), 15. Transport pathway (Categorical: Maternal inpatient transport, Maternal outpatient referral, Neonatal transport, Without referral) |
| **Neonatal factor** | 1. Gestational age (Double), 2. Male sex (Boolean), 3. Apgar score at 1 min. (Integer), 4. Apgar score at 5 min. (Integer), 5. Oxygen administration in the delivery room (Boolean), 6. Intubation in the delivery room (Boolean), 7. Umbilical cord milking or delayed cord clamping (Boolean), 8. Weight at birth (Integer), 9. Length at birth (Double), 10. Head circumference at birth (Double), 11. Z-value of weight at birth (Double), 12. Z-value of length at birth (Double), 13. Z-value of head circumference at birth (Double), 14. Respiratory distress syndrome (Boolean), 15. Persistent pulmonary hypertension of the newborn (Boolean), 16. Hypoxic-ischemic encephalopathy (Boolean) |
| **Facility factor** | 1. Facility level (Categorical: Tertiary, Secondary, Primary), 2. Facility provider (Categorical: Public hospital, Private hospital, Independent administrative agency, National hospital, Other hospital), 3. Annual admission of very low birth weight infants (Integer), 4. Annual admission of extremely low birth weight infants (Integer), 5. Number of beds for neonates (Integer), 6. Number of beds in neonatal intensive care unit (Integer), 7. Number of beds in maternal-fetal intensive care unit (Integer), 8. Headcount of neonatologists (Integer), 9. Headcount of nurses (Integer), 10. Availability of psychologists (Boolean), 11. Availability of pediatric surgery (Boolean), 12. Availability of cardiac surgery (Boolean), 13. Availability of neurosurgery (Boolean), 14. Availability of ophthalmologists (Boolean), 15. Availability of a follow-up system (Boolean) |
| **Auxiliary variables in multiple imputation (46 variables)** | |
| **Maternal factor** | 1. Chorioamnionitis (Boolean), 2. Grade of chorioamnionitis (Categorical: Without chorioamnionitis, Grade I, Grade II, Grade III) |
| **Neonatal factor** | 1. Air leak (Boolean), 2. Pulmonary hemorrhage (Boolean), 3. Days on oxygen (Integer), 4. Days on continuous positive airway pressure (Integer), 5. Days on mechanical ventilation (Integer), 6. Use of high-frequency oscillatory ventilation (Boolean), 7. Total dose of surfactant administration (Integer), 8. Days on nitric oxide (Integer), 9. Chronic lung disease at 28 days (Boolean), 10. Type of bronchopulmonary dysplasia (Categorical: Without BPD, I, II, III, III´, IV, V, VI), 11. Fraction of inspired oxygen at 36 weeks (Double), 12. Patent ductus arteriosus (PDA) (Boolean), 13. Indomethacin administration for PDA (Categorical: Without administration, Prophylactic use, Therapeutic use), 14. Surgical treatment of PDA (Boolean), 15. Late-onset circulatory collapse (Boolean), 16. Neonatal seizure (Boolean), 17. Intraventricular Hemorrhage (IVH) (Boolean), 18. Grade of IVH (Categorical: Without IVH, Grade I, Grade II, Grade III, Grade IV), 19. IVH with hydrocephalus (Boolean), 20. Periventricular leukomalacia (Boolean), 21. Intrauterine infection (Boolean), 22. Neonatal sepsis (Boolean), 23. Early-onset neonatal sepsis (Boolean), 24. Antibiotic administration (Boolean), 25. Hyperalimentation (Boolean), 26. Necrotizing enterocolitis (Boolean), 27. Focal intestinal perforation (Boolean), 28. Stage of retinopathy of prematurity (ROP) (Categorical: Stage 0–1, Stage 2, Stage 3, Stage 4–5 or APROP), 29. Treatment for ROP (Boolean), 30. Major malformation (Boolean), 31. Surgery for major malformation (Boolean), 32. Day of full feeding ($\geq$100 mL/kg/day) (Integer), 33. Red blood cell transfusion (Boolean), 34. Erythropoietin administration (Boolean), 35. Day of discharge (Integer), 36. Place of discharge (Categorical: Home, Foster home, General pediatric ward, Other hospital, Death), 37. Home oxygen therapy (Boolean), 38. Tracheostomy (Boolean), 39. Weight at discharge (Integer), 40. Length at discharge (Double), 41. Head circumference at discharge (Double), 42. Motor impairment (Boolean), 43. Visual impairment (Boolean), 44. Hearing impairment (Boolean) |

favored approach in the realm of medical real-world data research, where the dose-response relationship is not self-evident and intricate inter-feature dynamics are a common challenge [27]. The GBDT model ensures interpretability with the aid of the Shapley additive explanations (SHAP) values for tree-based machine learning models [28, 29]. We implemented GBDT models with the CatBoost module (version 1.0.6) of Python (version 3.10.8) [30].

Hyperparameter tuning was performed on each imputed "training and validation" set through five-fold cross-validation, according to the black-box optimization built on the tree-structured Parzen estimator algorithm using the optuna module (version 2.10.1) of Python [31]. Subsequently, a stratified 8:2 random allocation was employed to split this dataset into the training set and the validation set. We trained the GBDT-based models with the training set, applying early stopping based on the validation set to prevent overfitting.

## Performance evaluation

The temporal validity of discrimination, precision, and calibration was assessed by calculating the area under the receiver operating characteristic curves (AUROCs), the area under the precision-recall curves (AUPRCs), and the expected calibration errors (ECEs) for each of the 20 imputed test sets. Their medians and 95% confidence intervals (CIs) were then determined. To compare predictive performance on the test set, we built the multivariable logistic regression model based on the imputed "training and validation" set.

## SHAP analysis

SHAP values, rooted in Shapley values in game theory, quantify the impact of each feature on the prediction for each case [28, 29]. Similar to the coefficients in logistic regression based on categorical variables, the predicted probability is equal to the logistic transformation applied to the sum of SHAP values for each case. Positive values indicate that they made the prediction more pessimistic, and negative values indicate that they made the prediction more optimistic. A larger absolute value of the SHAP value indicates a greater impact on the prediction. Using the SHAP module (version 0.41.0) in Python, we embedded the test set features into SHAP values. We identified the 12 most influential predictors and illustrated their impact using dependence plots. Furthermore, we applied t-distributed stochastic neighbor embedding (t-SNE) to the high-dimensional vectors of SHAP values of individual cases of the first imputed test set to perform dimensionality reduction and verified the retention of critical information for prediction (perplexity = 30.0, scikit-learn version 1.1.3) [32]. To execute case clustering, we also conducted $k$-means clustering of SHAP values of the first imputed test set [33]. The number of clusters was determined using the elbow method [34]. The obtained clusters were visualized using the result of t-SNE.

## Ethics

This study was approved by the Research Ethics Committee of the University of Tokyo, Tokyo, Japan (registration number: 2021029NI-(1)). For the NRNJ database research, written informed consent, including secondary use, was obtained from the parents or guardians of all infants.

## Results

Out of 41,237 admissions of VPT and VLBW infants, cases admitted after the first 48 h of life, cases transferred during the acute phase, and those with missing outcomes were excluded (**Fig 1**). This resulted in 39,488, 39,096 and 40,291 infants for the prediction of "death or BPD", "death or severe BPD", and "death before discharge", respectively. **Table 2** summarizes the basic characteristics of the infants. No variable in Table 1 had missing rates of more than 30%; thus, omissions of variables were avoided. **S1 Table** displays the univariable and multivariable logistic regression for predicting "death or BPD", conducted as a complete case analysis. The

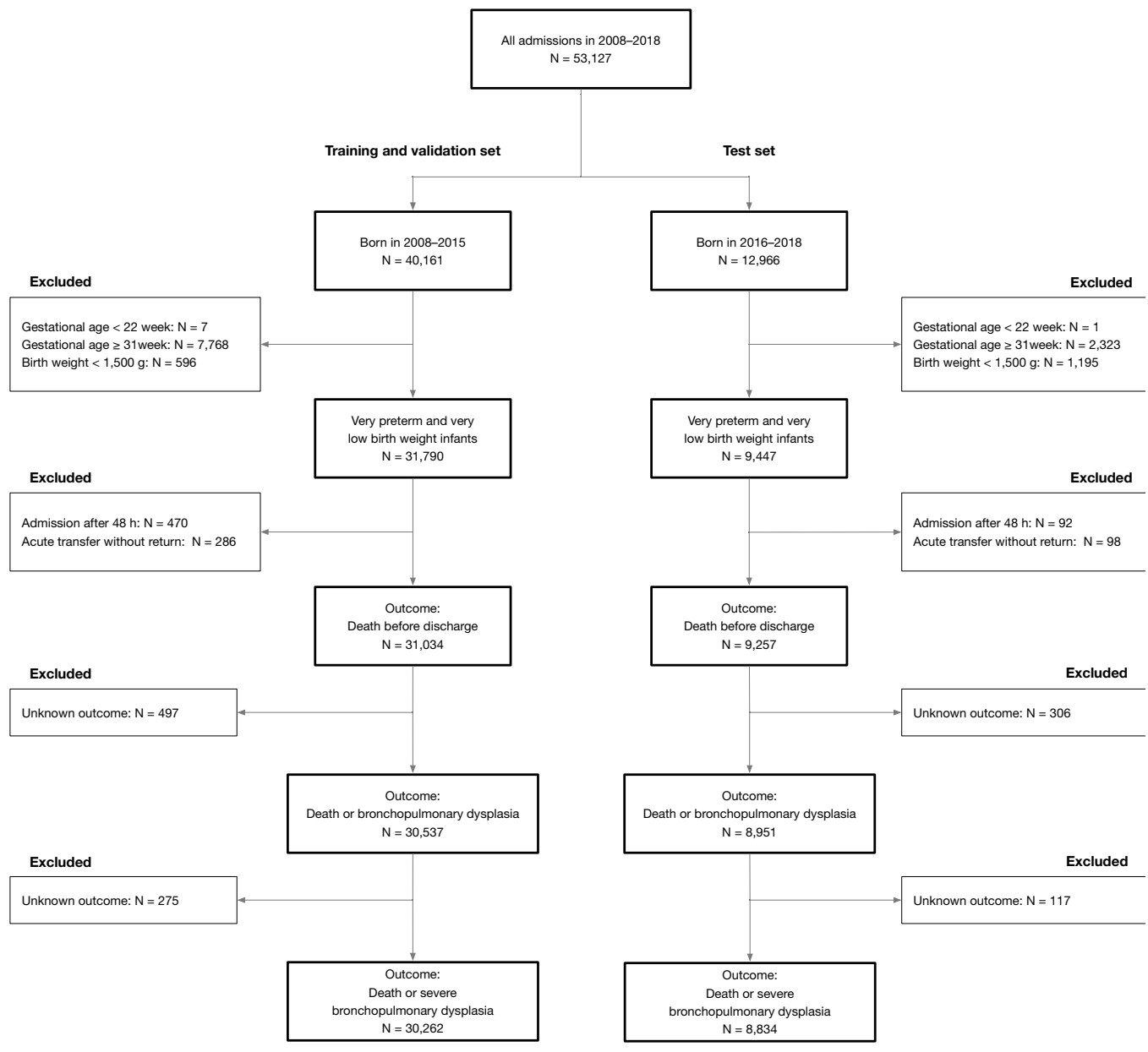

**Fig 1. Flowchart of patient selection and data segregation.**

degrees-of-freedom-adjusted generalized variance inflation factors for all variables were below 10.

In the prediction of death or BPD, the GBDT model demonstrated superior discrimination with a median AUROC of 0.828 (95% CI: 0.828–0.828), compared to 0.820 (95% CI: 0.820–0.821) for multivariable logistic regression based on multiple imputation (**Table 3** and **S2 Fig**). The GBDT model demonstrated superiority in precision with a median AUPRC of 0.766 (0.765–0.766) and in calibration with a median ECE of 0.0118 (95% CI: 0.0114–0.0123). The GBDT model also achieved a better median AUROC of 0.887 (95% CI: 0.887–0.888) and AUPRC of 0.403 (95% CI: 0.402–0.406) in mortality prediction (**S3 Fig**).

**Table 2. Basic characteristics.**

| Characteristic | Training and Validation Set[1], N = 30,537 | Test Set,[1] N = 8,951 |
| --- | --- | --- |
| **Death or bronchopulmonary dysplasia** | 11,167 (37%) | 3,518 (39%) |
| **Death or severe bronchopulmonary dysplasia** | 3,280 (11%) | 837 (9.5%) |
| (Missing) | 275 | 117 |
| **Death before discharge** | 2,079 (6.8%) | 593 (6.6%) |
| **Birth year** | | |
| (2008 | 3,235 (11%) | 0 (0%) |
| (2009 | 3,429 (11%) | 0 (0%) |
| (2010 | 4,003 (13%) | 0 (0%) |
| (2011 | 4,249 (14%) | 0 (0%) |
| (2012 | 4,231 (14%) | 0 (0%) |
| (2013 | 3,988 (13%) | 0 (0%) |
| (2014 | 3,734 (12%) | 0 (0%) |
| (2015 | 3,668 (12%) | 0 (0%) |
| (2016 | 0 (0%) | 3,345 (37%) |
| (2017 | 0 (0%) | 2,919 (33%) |
| (2018 | 0 (0%) | 2,687 (30%) |
| **Maternal age** | 32 (28, 36) | 33 (29, 36) |
| ((Missing) | 531 | 81 |
| **Gravidity** | 2.00 (1.00, 3.00) | 2.00 (1.00, 3.00) |
| ((Missing) | 432 | 126 |
| **Parity** | 0 (0, 1) | 0 (0, 1) |
| ((Missing) | 313 | 109 |
| **Number of fetuses** | | |
| 1 | 23,930 (78%) | 6,957 (78%) |
| 2 | 5,853 (19%) | 1,780 (20%) |
| 3 | 719 (2.4%) | 201 (2.2%) |
| 4 | 30 (<0.1%) | 13 (0.1%) |
| 5 | 5 (<0.1%) | 0 (0%) |
| **Birth order** | | |
| 1 | 27,136 (89%) | 7,922 (89%) |
| 2 | 3,148 (10%) | 959 (11%) |
| 3 | 244 (0.8%) | 67 (0.7%) |
| 4 | 8 (<0.1%) | 3 (<0.1%) |
| 5 | 1 (<0.1%) | 0 (0%) |
| **Monochorionic multiple** | 2,915 (9.6%) | 855 (9.6%) |
| (Missing) | 190 | 52 |
| **Maternal diabetes** | 1,038 (3.5%) | 578 (6.7%) |
| (Missing) | 476 | 334 |
| **Hypertensive disorders of pregnancy** | 5,599 (19%) | 1,546 (18%) |
| (Missing) | 347 | 256 |
| **Clinical chorioamnionitis** | 5,949 (20%) | 1,614 (19%) |
| (Missing) | 821 | 275 |
| **Premature rupture of membranes** | 9,840 (32%) | 2,892 (33%) |
| (Missing) | 223 | 181 |
| **Antenatal steroid administration** | 17,279 (57%) | 5,849 (67%) |
| (Missing) | 348 | 231 |
| **Non-reassuring fetal status** | 7,482 (25%) | 2,275 (26%) |

*(Continued)*

**Table 2.** (Continued)

| Characteristic | Training and Validation Set[1], N = 30,537 | Test Set,[1] N = 8,951 |
| --- | --- | --- |
| (Missing) | 470 | 189 |
| **Cephalic presentation** | 19,909 (66%) | 5,468 (63%) |
| (Missing) | 358 | 234 |
| **Delivery mode** | | |
| Caesarean section | 23,658 (78%) | 7,037 (80%) |
| Natural vaginal delivery | 6,493 (21%) | 1,655 (19%) |
| Forceps or vacuum delivery | 158 (0.5%) | 101 (1.1%) |
| (Missing) | 228 | 158 |
| **Transport pathway** | | |
| Maternal inpatient transport | 17,244 (57%) | 4,500 (52%) |
| Maternal outpatient referral | 7,991 (27%) | 2,905 (34%) |
| Neonatal transport | 1,424 (4.7%) | 414 (4.8%) |
| Without referral | 3,459 (11%) | 783 (9.1%) |
| (Missing) | 419 | 349 |
| **Gestational age** | 28.14 (26.00, 30.00) | 28.14 (26.00, 30.00) |
| **Male sex** | 15,969 (52%) | 4,693 (52%) |
| **Apgar score at 1 min.** | 5 (3, 7) | 5 (3, 7) |
| (Missing) | 374 | 87 |
| **Apgar score at 5 min.** | 8 (6, 9) | 7 (6, 8) |
| (Missing) | 542 | 93 |
| **Oxygen administration in the delivery room** | 26,836 (88%) | 7,949 (89%) |
| **Intubation in the delivery room** | 20,205 (66%) | 6,140 (69%) |
| **Umbilical cord milking or delayed cord clamping** | 7,442 (25%) | 2,702 (31%) |
| (Missing) | 928 | 344 |
| **Weight at birth** | 980 (731, 1,229) | 962 (710, 1,220) |
| **Length at birth** | 35.0 (32.0, 37.6) | 35.0 (31.5, 37.5) |
| (Missing) | 1,154 | 230 |
| **Head circumference at birth** | 25.40 (23.00, 27.00) | 25.10 (23.00, 27.00) |
| (Missing) | 1,690 | 294 |
| **Z-value of weight at birth** | -0.63 (-1.53, 0.04) | -0.66 (-1.59, 0.01) |
| **Z-value of length at birth** | -0.56 (-1.40, 0.18) | -0.55 (-1.38, 0.15) |
| (Missing) | 1,154 | 230 |
| **Z-value of head circumference at birth** | -0.10 (-0.71, 0.46) | -0.15 (-0.76, 0.42) |
| (Missing) | 1,690 | 294 |
| **Respiratory distress syndrome** | 20,706 (69%) | 6,171 (71%) |
| (Missing) | 415 | 229 |
| **Persistent pulmonary hypertension of the newborn** | 1,961 (6.5%) | 718 (8.3%) |
| (Missing) | 543 | 330 |
| **Hypoxic-ischemic encephalopathy** | 320 (1.1%) | 78 (0.9%) |
| (Missing) | 631 | 312 |
| **Facility level** | | |
| Tertiary | 23,912 (78%) | 7,049 (79%) |
| Secondary | 6,050 (20%) | 1,847 (21%) |
| Primary | 573 (1.9%) | 22 (0.2%) |
| (Missing) | 2 | 33 |
| **Facility provider** | | |
| Public hospital | 10,870 (39%) | 2,559 (34%) |

*(Continued)*

**Table 2.** (Continued)

| Characteristic | Training and Validation Set[1], N = 30,537 | Test Set,[1] N = 8,951 |
|---|---|---|
| Private hospital | 8,859 (32%) | 2,368 (31%) |
| Independent administrative agency | 6,322 (23%) | 2,072 (28%) |
| National hospital | 1,493 (5.4%) | 527 (7.0%) |
| Others | 0 (0%) | 0 (0%) |
| (Missing) | 2,993 | 1,425 |
| Annual admission of very low birth weight infants | 50 (34, 66) | 42 (32, 54) |
| Annual admission of extremely low birth weight infants | 22 (14, 32) | 21 (13, 28) |
| Number of beds for neonates | 33 (26, 42) | 33 (27, 41) |
| (Missing) | 9 | 0 |
| Number of beds in neonatal intensive care unit | 12 (9, 15) | 12 (9, 18) |
| Number of beds in maternal-fetal intensive care unit | 6 (3, 7) | 6 (3, 9) |
| (Missing) | 30 | 0 |
| Headcount of neonatologists | 7 (5, 8) | 8 (6, 10) |
| Headcount of nurses | 47 (39, 61) | 50 (41, 68) |
| (Missing) | 9 | 0 |
| Availability of psychologists | 22,340 (73%) | 7,912 (88%) |
| Availability of pediatric surgery | 25,308 (83%) | 7,343 (82%) |
| Availability of cardiac surgery | 15,373 (50%) | 4,592 (51%) |
| Availability of neurosurgery | 23,812 (78%) | 6,287 (70%) |
| Availability of ophthalmologists | 29,643 (97%) | 8,641 (97%) |
| Availability of a follow-up system | 24,784 (81%) | 7,958 (89%) |

[1]n (%); Median (IQR)

**Fig 2** shows the SHAP summary plots, which reveal the top 12 factors with the highest mean absolute SHAP values in each model, demonstrating their impact on the prediction. A SHAP summary plot is an arrangement of one-dimensional scatterplots of SHAP values created for each variable and displayed in order of the variables with the mean absolute SHAP values. The points on the right side represent contribution to the occurrence of the outcome,

**Table 3. Test performance of predictive models.**

| Model | AUROC*[†] | AUPRC*[‡] | ECE*[§] |
|---|---|---|---|
| **Outcome: Death or Bronchopulmonary Dysplasia** | | | |
| GBDT Model | 0.828 (0.828–0.828) | 0.766 (0.765–0.766) | 0.0118 (0.0114–0.0123) |
| Multivariable Logistic Regression Model | 0.820 (0.820–0.821) | 0.756 (0.755–0.756) | 0.0194 (0.0191–0.0197) |
| **Outcome: Death or Severe Bronchopulmonary Dysplasia** | | | |
| GBDT Model | 0.873 (0.873–0.873) | 0.457 (0.455–0.458) | 0.0318 (0.0317–0.0319) |
| Multivariable Logistic Regression Model | 0.861 (0.861–0.862) | 0.426 (0.423–0.427) | 0.0308 (0.0307–0.0310) |
| **Outcome: Death before Discharge** | | | |
| GBDT Model | 0.887 (0.887–0.888) | 0.403 (0.402–0.406) | 0.0310 (0.0310–0.0312) |
| Multivariable Logistic Regression Model | 0.876 (0.876–0.876) | 0.372 (0.369–0.374) | 0.0272 (0.0271–0.0275) |

*Median (95% confidence interval) of the 20 imputed test sets

[†]AUROC = area under the receiver operating characteristic curve

[‡]AUPRC = area under the precision-recall curve

[§]ECE = expected calibration error; GBDT = gradient boosting decision trees.

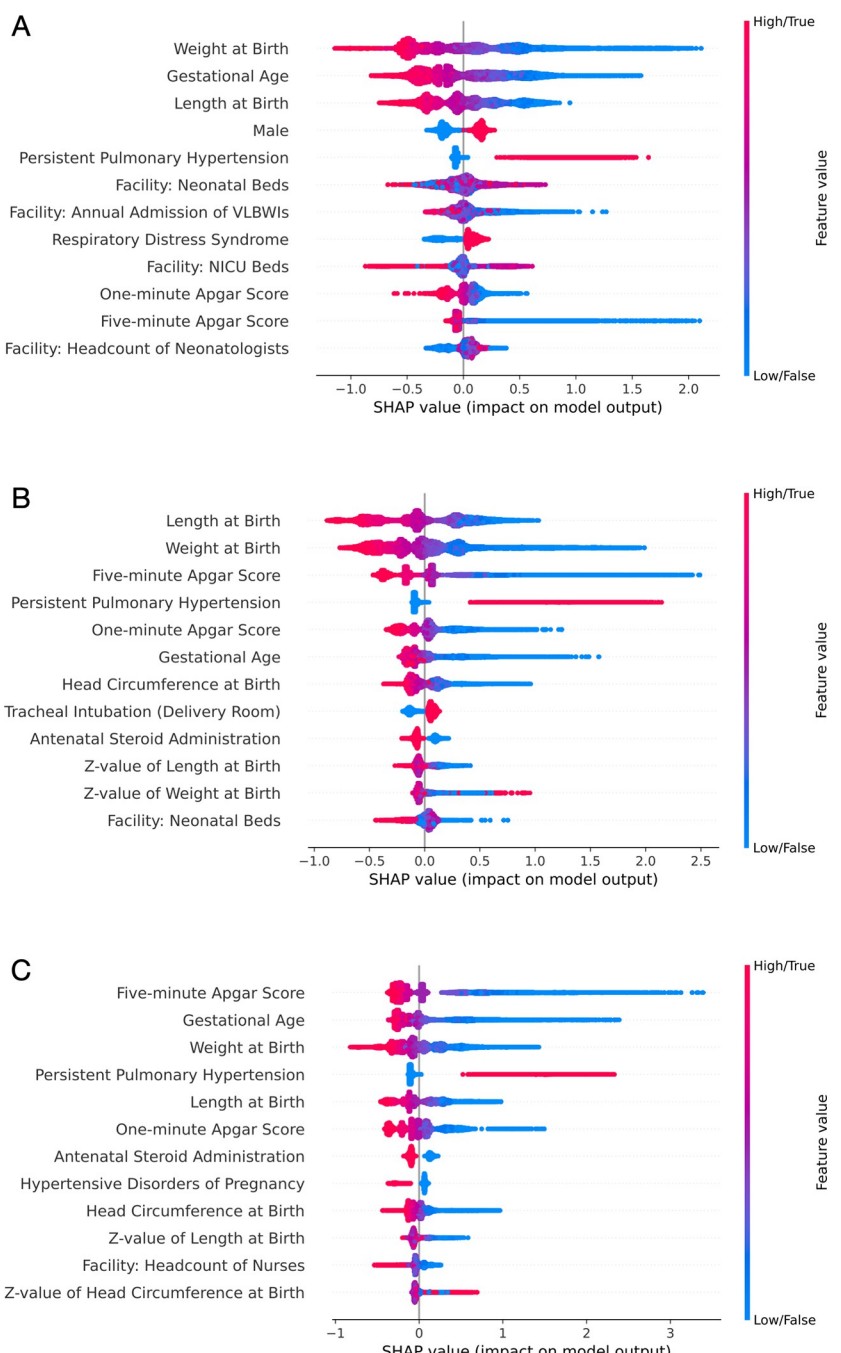

**Fig 2.** SHAP summary plots of the 20 most influential predictors (A) The predictive model for death or bronchopulmonary dysplasia (B) The predictive model for death or severe bronchopulmonary dysplasia (C) The predictive model for death before discharge.

while those found on the left side illustrate contribution to the non-occurrence of the outcome. The SHAP dependence plots, as illustrated in **S5–S7 Figs**, are two-dimensional scatter plots created for each of the 12 leading factors identified in the SHAP summary plots. Each SHAP dependency plot shows the dose-response relationship between the variable and its SHAP values as well as the interaction with the confounding factors used for color coding.

**Fig 2A** shows that unfavorable predictions for "death or BPD" were associated with lower weight at birth, lower gestational age at birth, shorter length at birth, male sex, PPHN, etc. Although the number of neonatal beds and neonatologists also influenced the prediction, the association with SHAP values was not distinct. **S5 Fig** shows that SHAP values consistently decrease with increasing birth weight and gestational weeks; moreover, it also indicates pessimistic predictions for facilities with fewer than 25 annual VLBW infants.

**Fig 2B** demonstrates that poor predictions for "death or severe BPD" were linked with shorter length at birth, lower weight at birth, lower 5-minute Apgar score, PPHN, lower 1-minute Apgar score, etc. **S6 Fig** reveals that the risk consistently decreases with an increase in birth weight; however, it also shows that the benefit of increased head circumference levels off at 40 cm, and gains from increased weeks of gestation plateau at 26 weeks of gestation. The influence of birth length was mitigated by the presence of PPHN. Furthermore, facilities with fewer than 20 neonatal beds were also identified as a risk factor for "death or severe BPD".

Adverse predictions for "death before discharge" were connected with lower 5-minute Apgar score, gestational age, weight at birth, PPHN, shorter length at birth, etc. (**Fig 2C**). Antenatal steroid administration and hypertensive disorders of pregnancy (HDP) were associated with lower SHAP values. **S7 Fig** illustrates that the decrease in predicted mortality associated with the higher 5-minute Apgar score stabilized at around 6. The dose-response relationships of gestational age at birth, birth weight, and birth length with their SHAP values were similar to those observed in predictive models for "death or severe BPD". The higher the number of full-time nurses in the neonatal ward, the more optimistic the predictions for mortality in extremely preterm infants became.

**S8–S10 Figs** show the result of t-SNE color-coded with the 12 most influential predictors for "death or BPD", "death or severe BPD", and "death before discharge", respectively. In these figures, a gradient based on gestational age and birth sizes was formed, with PPHN forming a distinct bunch. The high-volume facilities in the death or BPD prediction and HDP in mortality prediction also formed their own bunches. **Fig 3** illustrates each of the four clusters obtained by *k*-means clustering of SHAP values of the first imputed test set, using the t-SNE results. The observation that the clusters yielded were comprised cases in close proximity in t-SNE indicates that each cluster was comprised cases with similar patterns of SHAP values. As detailed below, each cluster exhibited distinct clinical characteristics. **S2 Table** shows that *k*-means clustering based on the "death or BPD" prediction principally resulted in stratification by gestational age: Cluster 1 with relatively advanced gestational age, Clusters 2 and 3 with moderate gestational (Cluster 2 being without PPHN and Cluster 3 with PPHN), and Cluster 4 with extreme prematurity. **S3 Table** indicates that *k*-means clustering based on the "death or

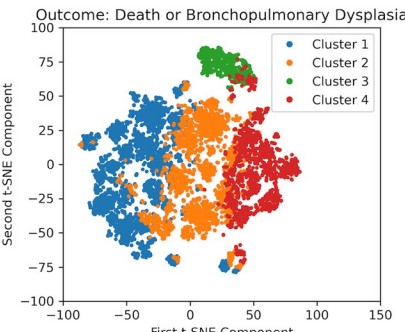 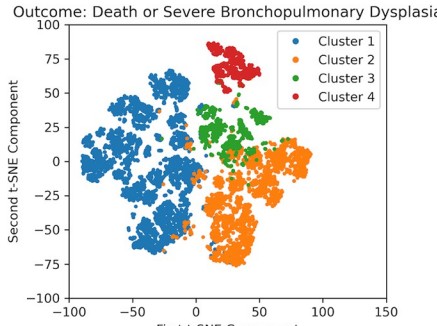 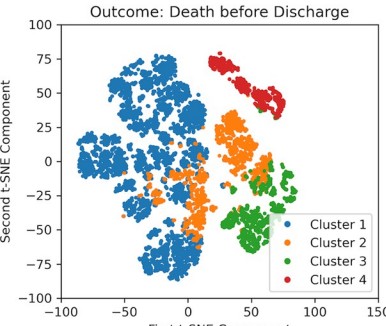

**Fig 3.** *k*-means clustering of the Shapley additive explanations values of the test set, visualized by t-distributed stochastic neighbor embedding. On the left are clusters based on the predictive model for death or bronchopulmonary dysplasia. In the middle, clusters are based on the predictive model for death or severe bronchopulmonary dysplasia. On the right, clusters are based on the predictive model for death before discharge.

severe BPD" prediction formed four distinct clusters corresponding to different clinical conditions: Cluster 1 with low mortality, Cluster 2 with extreme prematurity, Cluster 3 with low Apgar scores, and Cluster 4 with PPHN. In each cluster, pre-discharge death constituted over half of the composite outcome. **S4 Table** reveals that *k*-means clustering based on mortality prediction formed four distinct clusters based on highly similar characteristics as the "death or severe BPD" prediction.

## Discussion

Using only the clinical information available within the first 6 h postpartum, we developed GBDT models to predict "death or BPD," "death or severe BPD," and "death before discharge." These well-calibrated GBDT models outperformed multivariable logistic regression in terms of discrimination and precision. SHAP value analysis enabled the visualization of influential predictors and provided detailed insights into their effects on predictions. By applying *k*-means clustering to SHAP values, we achieved data-driven clustering that aligned with clinical insights for VPT and VLBW infants.

The GBDT model demonstrated superior test performance for each outcome, potentially attributed to the high expressive power of tree-based ensemble models. The lack of excessively poor performance, even with multivariable logistic regression models, may be attributed to the presence of linear-like relationships between the logit and variables, as indicated by SHAP dependence plots. Based on the results of the SHAP analysis, it can be inferred that the model effectively handled basic clinical information. Therefore, a breakthrough in discrimination capability may require the utilization of data from different modalities, such as new features obtained by lung ultrasound, processing of chest X-ray images, and time-series analysis of monitoring data. An underlying reason for the limited precision in mortality prediction could be the omission of life-threatening variables, such as chromosomal abnormalities and intraventricular hemorrhage, which are not always diagnosed immediately after birth. The collection of information related to fetal diagnosis and the timing of various complications could potentially improve precision.

The strong association between BPD prediction and prematurity, confirmed by SHAP analysis, is consistent with previous research [35, 36]. Continuous reduction in BPD risk with advancing gestational weeks (**S5 Fig**) coincides with the timing of alveolar differentiation and lung formation [37]. The substantial impact of birth size may reflect the consequence of fetal growth retardation [38, 39]. Studies of preterm infants intrinsically and inevitably involve conditioning by the occurrence of preterm birth, which acts as a collider for the risk of preterm birth such as non-reassuring fetal status, congenital diseases, infections, and placental abruption. Lower SHAP values in mortality prediction among infants with maternal HDP may be due to the reduced incidence of other preterm delivery risk factors, including unmeasured ones. The optimistic impact of antenatal steroid administration on mortality prediction may have been enhanced by the implication that the preterm delivery was not entirely unforeseeable. As for facility factors, fewer annual admissions of VLBW infants, neonatal beds, and full-time nurses were risk factors for "death or BPD", "death or severe BPD", and "death before discharge", respectively. Given that infants admitted to high-volume facilities are likely to have a higher severity of illness, these findings may suggest the importance of a skilled and sufficient workforce to improve outcomes. The inconsistent impact of the number of neonatal beds and neonatologists on the "death or BPD" prediction may suggest the presence of complex interactions with other predictive factors. The influence of facility-related factors may exhibit variations across countries, as the effect of facility volume on BPD did not yield statistical significance in prior research conducted in Germany [40].

The formation of clusters aligned with clinical significance through *k*-means clustering demonstrates the utility of SHAP values as vector representations of cases, thereby indicating the validity of our predictive models. In the *k*-means clustering of SHAP values derived from the "death or BPD" prediction, preterm infants with moderate gestational age were split according to PPHN, and a remarkable difference was observed in pre-discharge mortality (24% with PPHN vs. 3.2% without PPHN), compared with the incidence of survival discharge with BPD (47% vs. 38%) (**S2 Table**). This suggests that PPHN is more indicative of pre-discharge mortality than of the development of BPD. Consistently, Cluster 4 of the mortality prediction (**S4 Table**), characterized by PPHN, exhibited the highest mortality. The formation of distinct clusters derived from mortality prediction, including extreme prematurity, neonatal asphyxia, and PPHN, indicated the data-driven extraction of crucial features by the GBDT algorithm. Over half of the infants with the composite outcomes of death or severe BPD died before discharge, and the results of *k*-means clustering based on this outcome closely resembled that driven by the prediction of death before discharge (**S3 Table**). Survival with severe BPD was most prevalent in Cluster 2, supporting the importance of preventive interventions for highly premature infants.

## Limitations

This study had several limitations. First, owing to case selection and missing data, this retrospective cohort study based on a nationwide case registry may have had incomplete information. During the study period Japan recorded 64,896 births of VPT and VLBW infants, suggesting that over 60% of these infants were included in this study [41]. The chosen prognostic factors demonstrated missing data rates of less than 30% each. Missing data were imputed with the multiple imputation method. Second, though the temporal validity may imply a favorable performance of GBDT models trained on the relevant population, the external validity of our models remains unverified. We incorporated treatment options such as antenatal steroid administration based on the assumption that the treatment strategies had been standardized according to the established clinical guidelines. The prediction of neonatal outcomes is inevitably influenced by obstetric decisions such as cesarean sections and labor induction. Given that both the indications and the consequences of the treatments influence predictions, this may result in limited external validity. Third, established risk factors for BPD, including neonatal sepsis, symptomatic patent ductus arteriosus, and course of respiratory management, were not considered among the predictors owing to their unavailability immediately after birth [12, 42, 43]. Nevertheless, our GBDT models, which relied solely on early accessible predictors, demonstrated reasonable predictive performance. Lastly, inter-rater disagreement in clinical diagnosis, such as RDS and PPHN, can occur. It should be noted that this can affect the performance of the predictive models.

## Conclusions

We developed GBDT-based predictive models for BPD and mortality in VPT and VLBW infants using the NRNJ database. These models outperformed the multivariable logistic regression models. The SHAP analysis validated the appropriate response to the features. SHAP values generated clusters consistent with clinical insights. Therefore, our findings suggest that GBDT-based machine learning approaches may aid in the early postnatal prognosis prediction of BPD.

## Supporting information

**S1 Fig. Data flow and overview of the experiment.**
(PDF)

**S2 Fig. Test performance of the predictive models for death or bronchopulmonary dysplasia.** The upper panel shows the gradient boosting decision trees model, and the lower panel shows the multivariable logistic regression model. For each model, receiver operating characteristic curves, precision-recall curves, calibration plots, and histograms of predicted values are aligned from left to right.
(PDF)

**S3 Fig. Test performance of the predictive models for death or severe bronchopulmonary dysplasia.** The upper panel shows the gradient boosting decision trees model, and the lower panel shows the multivariable logistic regression model. For each model, receiver operating characteristic curves, precision-recall curves, calibration plots, and histograms of predicted values are aligned from left to right.
(PDF)

**S4 Fig. Test performance of the predictive models for death before discharge.** The upper panel shows the gradient boosting decision trees model, and the lower panel shows the multivariable logistic regression model. For each model, receiver operating characteristic curves, precision-recall curves, calibration plots, and histograms of predicted values are aligned from left to right.
(PDF)

**S5 Fig. SHAP dependence plots of the 12 most influential predictors of death or bronchopulmonary dysplasia.** Derived from the SHAP values of the 20 imputed test sets.
(PDF)

**S6 Fig. SHAP dependence plots of the 12 most influential predictors of death or severe bronchopulmonary dysplasia.** Derived from the SHAP values of the 20 imputed test sets.
(PDF)

**S7 Fig. SHAP dependence plots of the 12 most influential predictors of death before discharge.** Derived from the SHAP values of the 20 imputed test sets.
(PDF)

**S8 Fig. Distribution of the 12 most influential predictors of death or bronchopulmonary dysplasia, visualized by t-distributed stochastic neighbor embedding.** Derived from the SHAP values of the first imputed test set.
(PDF)

**S9 Fig. Distribution of the 12 most influential predictors of death or severe bronchopulmonary dysplasia, visualized by t-distributed stochastic neighbor embedding.** Derived from the SHAP values of the first imputed test set.
(PDF)

**S10 Fig. Distribution of the 12 most influential predictors of death before discharge, visualized by t-distributed stochastic neighbor embedding.** Derived from the SHAP values of the first imputed test set.
(PDF)

**S1 Table. Univariable and multivariable logistic regression for death or bronchopulmonary dysplasia.**
(DOCX)

**S2 Table. Characteristics of clusters based on prediction for death or bronchopulmonary dysplasia.**
(DOCX)

**S3 Table. Characteristics of clusters based on prediction for death or severe bronchopulmonary dysplasia.**
(DOCX)

**S4 Table. Characteristics of clusters based on prediction for death before discharge.**
(DOCX)

## Acknowledgments

We express our profound gratitude to the research collaborators and support staff at all participating facilities in the NRNJ database study for their unwavering commitment and invaluable contributions. The site investigators of the Neonatal Research Network of Japan (represented by Satoshi Kusuda, Kusuda-satoshi@nrnj.org) were as follows: Takashi Nasu (Obihiro Kosei Hospital); Ayumu Noro (JCHO Hokkaido Hospital); Toshihiko Mori (NTT East Sapporo Hospital); Ken Nagaya (Asahikawa Medical University); Masaru Shirai (Asahikawa Kosei Hospital); Yosuke Kaneshi (Kushiro Red Cross Hospital); Masaki Kobayashi (Sapporo Prefecture Medical University); Masato Mizushima (Sapporo City Hospital); Nobuhiro Takahashi (Tenshi Hospital); Yusuke Ohkado (Tomakomai Chity Hospital); Tatsuro Satomi (Nikko Kinen Hospital); Mika Nakajima (Hakodate Central Hospital); Tomofumi Ikeda (Aomori Prefecture Central Hospital); Genichiro Sotodate (Iwate Medical University); Takahide Hosokawa (Iwate Prefecture Ninohe Hospital); Masatoshi Sanjo (Sendai Red Cross Hospital); Takushi Hanita (Tohoku University); Hirokazu Arai (Akita Red Cross Hospital); Masato Ito (Akita University); Satoshi Watanabe (Yamagata Prefecture Central Hospital); Hiroshi Yoshida (Tsuruoka City Shonai Hospital); Tsutomu Ishii (National Fukushima Hospital); Maki Sato (Fukusima Prefecture Medical University); Yoshiya Yukitake (Ibaraki Children's Hospital); Yayoi Miyazono (Tsukuba University); Goro Asada (Tsuchiura Kyodo Hospital); Yumi Kono (Jichi Medical University); Yasuaki Kobayashi (Ashikaga Red Cross Hospital); Yasushi Oki (Kiryu Kosei General Hospital); Kenji Ichinomiya (Gunma Prefecture Children's Hospital); Toru Fujiu (Gunma University); Hideaki Fukushima (Ohta General Hospital); Tetsuya Kunikata (Saitama Medical University); Kazuhiko Kabe (Saitama Medical University Medical Center); Masaki Shimizu (Saitama Prefecture Children's Hospital); Chika Morioka (Kawaguchi City Medical Center); Motoichiro Sakurai (Kameda General Hospital); Naoto Nishizaki (Juntendo University Urayasu Hospital); Satoshi Toishi (Narita Red Cross Hospital); Harumi Otsuka (Chiba City Kaihin Hospital); Nozomi Ishii (Aiiku Hospital); Kenichiro Hosoi (Kyorin Univesity); Keiji Goishi (National International Medical Center); Yuji Ito (National Center for Child Health and Development); Hiromichi Shoji (Juntendo University); Atsuo Miyazawa (Showa University); Naoki Ito (Teikyo University); Ken Masunaga (Tokyo Metropolitan Otsuka Hospital); Reiko Kushima (Tokyo Metropolitan Bokuto Hospital); Sakae Kumasaka (Tokyo Katsushika Red Cross Perinatal Center); Manabu Sugie (Tokyo Medical and Dental University); Daisuke Haruhara (Tokyo Medical University); Satsuki Kakiuchi (Tokyo Women's Medical University); Riki Nishimura (Tokyo University); Kaoru Okazaki (Tokyo Metropolitan Children's Medical Center); Hitoshi Yoda (Toho University); Atsushi Nakao (Japan Red Cross Hospital); Ichiro Morioka (Nihon University); Daisuke Ogata (Yokohama City Hospital); Fumihiko Ishida (Yokohama City University Medical Center); Daisuke Nishi (Yokohama Rosai Hospital); Miho Sato (Yokosuka Kyosai Hospital); Ayako Fukuyama (Yokohama Medical Center); Kuriko Nakamura (Saiseikai Eastern Yokohama Hospital); Kanji Ogo (Odawara City Hospital); Masahiko Murase (Showa University Northern Yokohama Hospital); Katsuaki Toyoshima (Kanagawa Children's Medical Center); Isamu Hokuto (St. Marianna Medical

University); Maha Suzuki (St. Mariana Medical University Yokohama City Seibu Hospital); Atsushi Uchiyama (Tokai University); Yoshio Shima (Nippon Medical School Musashi Kosugi Hospital); Hidehiko Nakanishi (Kitasato University Hospital); Atsushi Nemoto (Yamanashi Prefecture Central Hospital); Tatsuya Yoda (Saku General Hospital); Yukihide Miyosawa (Shinshu University); Takehiko Hiroma (Nagano Children's Hospital); Yoshihisa Nagayama (Niigata City Hospital); Tohei Usuda (Niigata University); Rei Kobayashi (Nagaoka Red Cross Hospital); Takeshi Hutani (Toyama Prefectural Central Hospital); Taketoshi Yoshida (Toyama University); Shuya Nagaoki (Kanazawa University); Yasuhisa Ueno (Ishikawa Prefectural Central Hospital); Hiroshi Yamamoto (Gifu Prefectural Medical Center); Takeshi Arakawa (Gifu Prefecture Tajimi Hospital); Takashi Tachibana (Oogaki City Hospital); Tadayuki Kumagai (Yaizu City Hospital); Shigeru Oki (Seirei Hamamatsu Hospital); Reiji Nakano (Shizuoka Children's Hospital); Taizo Ueno (Shizuoka Saiseikai Hospital); Mitsuhiro Ito (Fujieda City Hospital); Akira Oishi (Hamamatsu Medical University); Hikaru Yamamoto (Toyota Memorial Hospital); Hiroshi Takeshita (Aichi Medical University); Yuichi Kato (Anjokosei Hospital); Kuniko Ieda (Koritsu Tosei Hospital); Koji Takemoto (Konankosei Hospital); Masashi Miyata (Fujita Medical University); Osamu Shinohara (Handa City Hospital); Yasunori Koyama (Toyohashi City Hospital); Osuke Iwata (Nogoya City University); Takahiro Muramatsu (Nagoya City Seibu Medical Cneter); Akinobu Taniguchi (Nagoya University); Makoto Ohshiro (Nagoya Red Cross Daiici Hospital); Masanori Kowaki (Nagoya Red Cross Daini Hospital); Hiroshi Uchizono (National Mie Central Medical Center); Takahide Yanagi (Shiga Medical University); Kenji Nakamura (Otsu Red Cross Hospital); Masahito Yamamoto (Nagahama Red Cross Hospital); Jitsuko Ohira (Uji Tokushukai Hospital); Ryosuke Araki (Kyoto University); Daisuke Kinoshita (Kyoto Red Cross Daiichi Hospital); Ryuji Hasegawa (Kyoto Prefecture Medical University); Hiroshi Komatsu (National Maizuru Medical Center); Shinsuke Adachi (Fukuchiyama City Hospital); Toru Yamakawa (Japan Baptist Hospital); Masahiko Kai (Bell Land General Hospital); Hiroshi Sumida (Rinku General Hospital); Hirotaka Minami (Takatsuski General Hospital); Kenji Mine (Kansai Medical University); Satoru Ogawa (Saiseikai Suita Hospital); Ryoko Yoshinare (Hannan Central Hospital); Kiyoaki Sumi (Aizenbashi Hospital); Akihiro Takatera (Chifune Hospital); Satoshi Onishi (Osaka Metropolitan University); Taho Kim (Osaka City Sumiyoshi Hospital); Hiroyuki Ichiba (Osaka City General Hospital); Misao Yoshii (Osaka Red Cross Hospital); Hitomi Okabe (Osaka University); Shinya Hirano (Osaka Women's and Children's Hospital); Makoto Nabetani (Yodogawa Christian Hospital); Masaaki Ueda (Toyooka General Hospital); Takahiro Okutani (Saiseikai Hyogo Hospital); Masaru Yamakawa (Kobe City Medical Center Central Hospital); Kazumichi Fujioka (Kobe University); Tomoaki Ioroi (Himeji Red Cross Hospital); Takeshi Utsunomiya (Hyogo Medical University Hospital); Seiji Yoshimoto (Kobe Children's Hospital); Tamaki Ohashi (Hyogo Prefectural Awaji Medical Center); Yoshinobu Nishida (Hyogo Prefectural Amagasaki General Hospital); Toshiya Nishikubo (Nara Prefecture Medical University); Ken Kumagaya (Wakayama Prefecture Medical University); Akiko Tamura (Tottori Prefectural Central Hospital); Masumi Miura (Tottori University); Yuki Hasegawa (Matsue Red Cross Hospital); Rie Kanai (Shimane Prefectural Central Hospital); Koichi Tsukamoto (Okayama University); Misao Kageyama (National Okayama Medical Center); Takashi Nakano (Kawasaki Medical University); Hironobu Tokumasu (Kurashiki Central Hospital); Rie Fukuhara (Hiroshima Prefectural Hospital); Yutaka Nishimura (Hiroshima City Central Hospital); Seiichi Hayakawa (Hiroshima University); Yasuhiko Sera (National Kure Medical Center); Masahiro Tahara (Tsuchiya General Hospital); Keiko Hasegawa (Yamaguchi Prefecture Medical Center); Kazumasa Takahashi (Yamaguchi University); Hiroshi Tateishi (Tokuyama Central Hospital); Tomomasa Terada (Tokushima Prefecture Central Hospital); Takahiko Saijo (Tokushima University); Toru Kuboi (Shikoku Medical Center for Children and Adults);

Shinosuke Akiyoshi (Ehime Prefectural Central Hospital); Yusei Nakata (Kochi Health Science Center); Hideaki Harada (Kurume University); Masayuki Ochiai (Kyushu University); Toshinori Nakashima (National Kokura Medical Center); Toshiharu Hikino (National Kyushu Medical Center); Shutaro Suga (University of Occupational and Environmental Health Japan); Mitsuaki Unno (Saint Maria Hospital); Hiroshi Kanda (Iizuka Hospital); Yasushi Takahata (Fukuoka City Children's Hospital); Hiroyasu Kawano (Fukuoka University); Takayuki Kokubo (Kitakyushu City Hospital); Toshimitsu Takayanagi (National Saga Hospital); Mikio Aoki (National Nagasaki Medical Center); Muneichiro Sumi (Sasebo City Hospital); Tsutomu Ogata (Nagasaki University); Kei Inomata (Kumamoto City Hospital); Masanori Iwai (Kumamoto University); Naoki Fukushima (Almeida Memorial Hospital); Koichi Iida (Oita Prefectural Hospital); Yuki Kodama (Miyazaki University); Yuko Maruyama (Imakyure General Hospital); Takuya Tokuhisa (Kagoshima City Hospital); Yoriko Kisato (Okinawa Prefectural Central Hospital); Tatsuo Oshiro (Okinawa Prefectural Nanbu Medical Center/Nanbu Child Medical Center); Kazuhiko Nakasone (Okinawa Red Cross Hospital); Asao Yara (Naha City Hospital);

We also thank the medical editors of Editage, for English editing and proofreading of our manuscript.

## Author Contributions

**Conceptualization:** Kota Yoneda.

**Formal analysis:** Kota Yoneda.

**Investigation:** Kota Yoneda.

**Methodology:** Kota Yoneda, Tomohisa Seki, Yoshimasa Kawazoe.

**Resources:** Kazuhiko Ohe.

**Software:** Kota Yoneda.

**Supervision:** Tomohisa Seki, Yoshimasa Kawazoe, Kazuhiko Ohe, Naoto Takahashi.

**Visualization:** Kota Yoneda.

**Writing – original draft:** Kota Yoneda.

**Writing – review & editing:** Tomohisa Seki, Yoshimasa Kawazoe, Kazuhiko Ohe, Naoto Takahashi.

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
