## [Decision Letter · Decision Letter 0]

20 Dec 2023

PONE-D-23-39831Immediate postnatal prediction of death or bronchopulmonary dysplasia among very preterm and very low birth weight infants based on gradient boosting decision trees algorithm: A nationwide database study in Japan.PLOS ONE

Dear Dr. Kawazoe,

Thank you for submitting your manuscript to PLOS ONE. After careful consideration, we feel that it has merit but does not fully meet PLOS ONE’s publication criteria as it currently stands. Therefore, we invite you to submit a revised version of the manuscript that addresses the points raised during the review process.

We look forward to receiving your revised manuscript.

Kind regards,

Kazumichi Fujioka

Academic Editor

PLOS ONE

Journal Requirements:

   "Yoshimasa Kawazoe belongs to the “Artificial Intelligence and Digital Twin Development in Healthcare, Graduate School of Medicine, The University of Tokyo” which is an endowment department. However, the sponsors had no influence over the interpretation, writing, or publication of this work."

4. In this instance it seems there may be acceptable restrictions in place that prevent the public sharing of your minimal data. However, in line with our goal of ensuring long-term data availability to all interested researchers, PLOS’ Data Policy states that authors cannot be the sole named individuals responsible for ensuring data access (http://journals.plos.org/plosone/s/data-availability#loc-acceptable-data-sharing-methods).

5. One of the noted authors is a group or consortium Neonatal Research Network of Japan. In addition to naming the author group, please list the individual authors and affiliations within this group in the acknowledgments section of your manuscript. Please also indicate clearly a lead author for this group along with a contact email address.

Additional Editor Comments:

**This is well conducted study, and I believe it is worth for publishing after minor revision.**

Reviewers' comments:

Reviewer's Responses to Questions

**Comments to the Author**

1. Is the manuscript technically sound, and do the data support the conclusions?

Reviewer #1: Yes

Reviewer #2: Yes

2. Has the statistical analysis been performed appropriately and rigorously? 

Reviewer #1: Yes

Reviewer #2: I Don't Know

3. Have the authors made all data underlying the findings in their manuscript fully available?

Reviewer #1: Yes

Reviewer #2: Yes

4. Is the manuscript presented in an intelligible fashion and written in standard English?

Reviewer #1: Yes

Reviewer #2: Yes

5. Review Comments to the Author

Reviewer #1: The authors have presented their predictive models for BPD and mortality based on immediate postnatal clinical data. The gradient boosting decision trees (GBDT) algorithm was adopted to predict BPD and mortality using predictors within the first 6 h postpartum. We assessed the temporal validity and evaluated model adequacy using Shapley additive explanations (SHAP) values. It is an interesting paper in terms of prediction modeling but a few clinical issues have to be addresses to validate the findings of the paper.

1. Please elaborate on why a prediction model for BPD of death is needed at an early postnatal age of 6hr. Reasons other that resource allocation might be participation in clinical trial, parental counselling, or clinical practice variations such as fluid restriction and more gentle ventilation techniques.

2. Cord blood transfusion is a vague term. Is this referring to delayed cord clamping or cord milking? Please change to a more universal terminology.

3. Please check the model to verify if it can predict the severity of BPD (mild/moderate/severe). If the patient is extremely immature (i.e. less than 24 weeks gestations) BPD will be inevitable but the severity will be helpful to know.

4. PPHN is an important parameter in your model. However in all births, pulmonary vessels are constricted in utero and relax after birth. Thus a degree of physiologic pulmonary hypertension is normal in all babies in the immediate postnatal period. PPHN could denote a delayed but normal transition after birth to a severe PPHN needing treatment with inhaled nitric oxide and other medications. The authors should specify PPHN by using an objective echocardiac measurement to define PPHN. It would be helpful if the treatment for PPHN were analyzed as well.

5. The precise definition for RDS and the treatment employed should be analyzed as well. Some papers use RDS treated with surfactant to denote the severity of the condition.

Reviewer #2: This study successfully developed a postnatal prediction model of death or bronchopulmonary dysplasia among very preterm and very low birth weight infants within 6 hours of birth with a nationwide study population. The study has been well conducted, and the interpretation of the results seems reasonable. However, I have several questions as below.

1. Introduction: The aim of a prediction model is unclear. Please introduce the background to explain the reason why a prediction model is needed in clinical settings. It isn't easy to understand the authors' motivation for developing a prediction model.

2. Discussion: In relation to question 1, please consider how this innovative model will impact future clinical practice. How important is it to predict BPD or death at birth immediately? That will be helpful for understanding the importance of this prediction model.

3. Results: I think that descriptions of Figures 2 and 3 are needed. What did the authors want to show using Figures 2 and 3? For example, higher weight, gestational age, and length at birth had a negative impact on SHAP values of BPD or death, and so on.

4. Table 3: In predicting death before discharge, ECE seems to be better in the multivariate logistic model than in the GBDT model.

5. L218, I don't think that the categorical variables are represented in gray.

6. Fig 2: How did the authors select 20 from 46 variables? I found the data of Figs S4 and S5, but several factors, including neonatal and NICU beds and headcount of neonatologists, seem not related to SHARP value positively or negatively. In contrast, HDP clearly negatively affected SHARP values, but I could not find any comments on the results, even in the discussion.

7. L223-224, 227-228: Please move the definition of variables from result to methods.

8. L231, S5.Fig. How were these variables evaluated as crucial predictors?

9. L243, S7.Fig: HDP seems to be forming a distinct branch. Why was it ignored?

10. In relation to question 6, please discuss the effect of HDP, antenatal corticosteroids, and the facility characteristics on the outcome, too.

6. PLOS authors have the option to publish the peer review history of their article (what does this mean?). If published, this will include your full peer review and any attached files.

Reviewer #1: No

Reviewer #2: No

---

## [Author Response · Author response to Decision Letter 0]

29 Jan 2024

January 29, 2024

Dr. Kazumichi Fujioka

Academic Editor

PLOS ONE

Dear Editor,

We greatly appreciate the time and effort you and the reviewers dedicated to assessing our paper critically and are pleased to resubmit the revised version of our manuscript to PLOS ONE. 

Your comments, as well as those of the reviewers, were highly insightful and enabled us to greatly improve the quality of our manuscript. The manuscript has been revised following careful consideration of the reviewers’ comments and suggestions.

In the following pages, we have provided our point-by-point response to all comments. We hope that you will find our revised manuscript suitable for publication in PLOS ONE. Please let us know if you require any further information. 

== Response to the Editor’s Comments == 

Answer: 

In accordance with your comment, we have rechecked the details and made some minor corrections as follows:

Page 1, Line 18:

E-mail: kawazoe@m.u-tokyo.ac.jp (YK)

Page 1, Line 20:

^ Membership of the Neonatal Research Network of Japan is provided in the Acknowledgements.

Page 2, Line 22:

Data availability

Page 7, Line 108:

Materials and methods

Answer: 

We utilized existing Python packages and R libraries in the usual way; hence, we do not believe that the condition of “author-generated code underpins the findings” applies in our case.

"Yoshimasa Kawazoe belongs to the “Artificial Intelligence and Digital Twin Development in Healthcare, Graduate School of Medicine, The University of Tokyo” which is an endowment department. However, the sponsors had no influence over the interpretation, writing, or publication of this work."

Answer: 

In accordance with your comment, we have added the relevant sentence to the cover letter and the “Conflict of Interest” statement of the manuscript.

4. In this instance it seems there may be acceptable restrictions in place that prevent the public sharing of your minimal data. However, in line with our goal of ensuring long-term data availability to all interested researchers, PLOS’ Data Policy states that authors cannot be the sole named individuals responsible for ensuring data access (http://journals.plos.org/plosone/s/data-availability#loc-acceptable-data-sharing-methods).

Answer: 

In accordance with your comment, we have added the contact information for the Office of Health Research Studies.

Page 1, Lines 24–29:

For data sharing of the dataset generated during this study, please contact the corresponding author or the Artificial Intelligence and Digital Twin in Healthcare, Graduate School of Medicine, The University of Tokyo, Japan (Email: aih-office@m.u-tokyo.ac.jp), or the Office for Human Research Studies, Graduate School of Medicine and Faculty of Medicine, The University of Tokyo, Faculty of Medicine Bldg. 2 4F, 7-3-1 Hongo, Bunkyo-ku, Tokyo 113-0033, Japan (Email: ethics@m.u-tokyo.ac.jp).

5. One of the noted authors is a group or consortium Neonatal Research Network of Japan. In addition to naming the author group, please list the individual authors and affiliations within this group in the acknowledgments section of your manuscript. Please also indicate clearly a lead author for this group along with a contact email address.

Answer: 

In accordance with your comment, we have listed the NRNJ members and their affiliations in the Acknowledgments section.

Answer: 

In accordance with your comment, we have included the ethics statement in the Ethics subsection of the Materials and Methods section.

Page 12, Lines 195–197 (Materials and Methods)

This study was approved by the Research Ethics Committee of the University of Tokyo, Tokyo, Japan (registration number: 2021029NI-(1)). For the NRNJ database research, written informed consent, including secondary use, was obtained from the parents or guardians of all infants.

Answer: 

No papers were retracted; hence, the list of references remains unchanged.

== Response to the Reviewers’ Comments == 

Reviewer #1

1. Please elaborate on why a prediction model for BPD of death is needed at an early postnatal age of 6hr. Reasons other that resource allocation might be participation in clinical trial, parental counselling, or clinical practice variations such as fluid restriction and more gentle ventilation techniques.

Answer: 

In light of your comment, we have added an explanation to the Introduction section regarding the importance and significance of our predictive models.

Page 5, Line 85–92:

Early postnatal prediction of preterm birth complications would enable the reduction of complications through alterations in treatment strategies given the evidence that advanced ventilators offering volume-targeted ventilation and/or non-invasive respiratory support can lower the risk of BPD [7,8]. The implementation of early risk assessments would not only allow for individualized treatments but also improve the feasibility of conducting clinical trials stratified by severity and enhance the quality of parental counseling at admission. However, previously reported predictive models pose challenges in performing early postnatal predictions of BPD due to multiple reasons.

2. Cord blood transfusion is a vague term. Is this referring to delayed cord clamping or cord milking? Please change to a more universal terminology.

Answer: 

Umbilical cord milking and delayed cord clamping were registered in the NRNJ database without distinction; that is, both were referred to as “umbilical cord blood transfusion” without distinction. In accordance with your comment, we have clarified the expressions as follows:

Lines 139 and 212 (Tables 1 and 2):

Umbilical cord milking or delayed cord clamping

3. Please check the model to verify if it can predict the severity of BPD (mild/moderate/severe). If the patient is extremely immature (i.e. less than 24 weeks gestations) BPD will be inevitable but the severity will be helpful to know.

Answer: 

Following your suggestion, we have additionally developed a predictive model for “death or severe BPD” to make the study more clinically useful (Figs 1–3; Table 3; Figs S3, 6, and 9; and Table S3). This model exhibited characteristics intermediate to those of the preceding two predictive models. The S3 Table highlighted the burden of severe BPD in extremely preterm infants.

We defined “death and severe BPD” as death before discharge or invasive mechanical ventilation at 36 weeks post-menstrual age, respectively, a definition that is sometimes referred to as type 2 severe BPD [1]. The old and new severity classifications by the National Institute of Child Health and Human Development (NICHD) could not be applied since nasal continuous positive airway pressure (CPAP) and high flow nasal cannula (HFNC) were not distinguished in the NRNJ database [2,3].

Page 7, Lines 128–129 (Materials and Methods)

We also created a prediction model for “death or severe BPD”, where “severe BPD” was defined as the need for mechanical ventilation at 36 weeks postmenstrual age.

Page 20, Lines 243–249 (Results)

Fig 2B demonstrates that poor predictions for death or severe BPD were linked with shorter length at birth, lower weight at birth, lower 5-minute Apgar score, PPHN, lower 1-minute Apgar score, etc. S6 Fig reveals that the risk consistently decreases with an increase in birth weight; however, it also shows that the benefit of increased head circumference levels off at 40 cm, and gains from increased weeks of gestation plateau at 26 weeks of gestation. The influence of birth length was mitigated by the presence of PPHN. Furthermore, facilities with fewer than 20 neonatal beds were also identified as a risk factor for “death or severe BPD”.

Pages 21, Lines 277–281 (Results)

S3 Table indicates that k-means clustering based on “death or severe BPD” prediction formed four distinct clusters corresponding to different clinical conditions: Cluster 1 with low mortality, Cluster 2 with extreme prematurity, Cluster 3 with low Apgar scores, and Cluster 4 with PPHN. In each cluster, pre-discharge death constituted over half of the composite outcome.

Page 25, Lines 345–349 (Discussion)

Over half of the infants with the composite outcomes of death or severe BPD died before discharge, and the results of k-means clustering based on this outcome closely resembled that driven by the prediction of death before discharge (S3 Table). Survival with severe BPD was most prevalent in Cluster 2, supporting the importance of preventive interventions for highly premature infants.

4. PPHN is an important parameter in your model. However in all births, pulmonary vessels are constricted in utero and relax after birth. Thus a degree of physiologic pulmonary hypertension is normal in all babies in the immediate postnatal period. PPHN could denote a delayed but normal transition after birth to a severe PPHN needing treatment with inhaled nitric oxide and other medications. The authors should specify PPHN by using an objective echocardiac measurement to define PPHN. It would be helpful if the treatment for PPHN were analyzed as well.

Answer: 

In the NRNJ database, PPHN was defined as “pulmonary hypertension and right-to-left shunt blood flow diagnosed by echocardiography in the acute phase.” In Japan, neonatologists perform repeated echocardiographic examinations as necessary and in a timely manner [4]. Data on specific measurements and detailed treatments were not available; thus, limiting further detailed analyses. We have documented concerns regarding inter-evaluator discrepancies in the study limitations.

Page 26, Lines 368–370:

Lastly, inter-rater disagreement in clinical diagnosis, such as RDS and PPHN, can occur. It should be noted that this can affect the performance of the predictive models.

5. The precise definition for RDS and the treatment employed should be analyzed as well. Some papers use RDS treated with surfactant to denote the severity of the condition.

Answer: 

In the NRNJ database, RDS was defined as a clinical diagnosis based on clinical symptoms, chest X-ray, and microbubble test. Data on specific measurements and surfactant administration in the acute phase were not available, hence limiting further detailed analyses. We have documented concerns regarding inter-evaluator discrepancies in the limitations.

Page 26, Lines 368–370:

Lastly, inter-rater disagreement in clinical diagnosis, such as RDS and PPHN, can occur. It should be noted that this can affect the performance of the predictive models.

Reviewer #2 

1. Introduction: The aim of a prediction model is unclear. Please introduce the background to explain the reason why a prediction model is needed in clinical settings. It isn't easy to understand the authors' motivation for developing a prediction model.

Answer: 

The primary motivation for early prediction is that if risks could be assessed immediately after birth, BPD could be reduced by modifying treatment strategies. In accordance with your comments, as well as those of Reviewer 1, we have added a clarification in the Introduction section regarding the utility of predictive models in clinical practice.

Page 6, Lines 88-95:

Early postnatal prediction of preterm birth complications would enable the reduction of complications through alterations in treatment strategies, given the evidence that advanced ventilators offering volume-targeted ventilation and/or non-invasive respiratory support can lower the risk of BPD [7,8]. The implementation of early risk assessments would not only allow for individualized treatments but also improve the feasibility of conducting clinical trials stratified by severity and enhance the quality of parental counseling at admission. However, previously reported predictive models pose challenges in performing early postnatal predictions of BPD due to multiple reasons.

2. Discussion: In relation to question 1, please consider how this innovative model will impact future clinical practice. How important is it to predict BPD or death at birth immediately? That will be helpful for understanding the importance of this prediction model.

Answer: 

This study is expected to contribute to the development of neonatal science not only through the realization of individualized medicine but also through clinical research according to the severity of the disease. We have described the future value of this study in the Introduction as follows:

Page 6, Lines 88-95:

Early postnatal prediction of preterm birth complications would enable the reduction of complications through alterations in treatment strategies, given the evidence that advanced ventilators offering volume-targeted ventilation and/or non-invasive respiratory support can lower the risk of BPD [7,8]. The implementation of early risk assessments would not only allow for individualized treatments but also improve the feasibility of conducting clinical trials stratified by severity and enhance the quality of parental counseling at admission. However, previously reported predictive models pose challenges in performing early postnatal predictions of BPD due to multiple reasons.

3. Results: I think that descriptions of Figures 2 and 3 are needed. What did the authors want to show using Figures 2 and 3? For example, higher weight, gestational age, and length at birth had a negative impact on SHAP values of BPD or death, and so on.I'm sorry, I don't see any text or context in your message. Please let me know how I can assist you.

Answer: 

In our initial manuscript, the explanation of Fig 2 was intertwined with the descriptions of S4 and S5 Figs, which might have caused some confusion. We have revised the manuscript to clearly distinguish the information that can be extracted solely from Fig 2. 

Fig 3, in conjunction with S2–4 Tables, was generated to illustrate the utility of SHAP values as vector representations for each case, thereby establishing the validity of this model. These explanations have been incorporated for further clarification. 

Pages 20-22, Lines 227-261 (Results)

Fig 2 shows the SHAP summary plots which reveal the top 12 factors with the highest mean absolute SHAP values in each model, demonstrating their impact on the prediction. A SHAP summary plot is an arrangement of one-dimensional scatterplots of SHAP values created for each variable and displayed in order of the variables with the mean absolute SHAP values. The points on the right side represent contribution to the occurrence of the outcome, while those found on the left side illustrate contribution to the non-occurrence of the outcome. The SHAP dependence plots, as illustrated in S5–7 Figs, are two-dimensional scatter plots created for each of the 12 leading factors identified in the SHAP summary plots. Each SHAP dependency plot shows the dose-response relationship between the variable and its SHAP values as well as the interaction with the confounding factors used for color coding.

Fig 2A shows that unfavorable predictions for “death or BPD” were associated with lower weight at birth, lower gestational age at birth, shorter length at birth, male sex, PPHN, etc. Although the number of neonatal beds and neonatologists also influenced the prediction, the association with SHAP values was not distinct. S5 Fig shows that SHAP values consistently decrease with increasing birth weight and gestational weeks; moreover, it also indicates pessimistic predictions for facilities with fewer than 25 annual VLBW infants.

Fig 2B demonstrates that poor predictions for “death or severe BPD” were linked with shorter length at birth, lower weight at birth, lower 5-minute Apgar score, PPHN, lower 1-minute Apgar score, etc. S6 Fig reveals that the risk consistently decreases with an increase in birth weight; however, it also shows that the benefit of increased head circumference levels off at 40 cm, and gains from increased weeks of gestation plateau at 26 weeks of gestation. The influence of birth length was mitigated by the presence of PPHN. Furthermore, facilities with fewer than 20 neonatal beds were also identified as a risk factor for “death or severe BPD”.

Adverse predictions for “death before discharge” were connected with lower 5-minute Apgar score, gestational age, weight at birth, PPHN, shorter length at birth, etc. (Fig 2C). Antenatal steroid administration and hypertensive disorders of pregnancy (HDP) were associated with lower SHAP values. S7 Fig illustrates that the decrease in predicted mortality associated with the higher 5-minute Apgar score stabilized at around 6. The dose-response relationships of gestational age at birth, birth weight, and birth length with their SHAP values were similar to those observed in predictive models for “death or severe BPD”. The higher the number of full-time nurses in the neonatal ward, the more optimistic the predictions for mortality in extremely preterm infants became.

Page 22, Lines 271-275 (Results)

Fig 3 illustrates each of the four clusters obtained by k-means clustering of SHAP values of the test set variables, using the t-SNE results. The observation that the yielded clusters comprised cases in close proximity in t-SNE indicates that each cluster comprised cases with similar patterns of SHAP values. As detailed below, each cluster exhibited distinct clinical characteristics.

Pages 25-26, Lines 337-339 (Discussion)

The formation of clusters aligned with clinical significance through k-means clustering demonstrates the utility of SHAP values as vector representations of cases, thereby indicating the validity of our predictive models.

4. Table 3: In predicting death before discharge, ECE seems to be better in the multivariate logistic model than in the GBDT model.

Answer: 

The GBDT models demonstrated superior predictive performance in terms of AUROC and AUPRC. The group with predicted probabilities in the 90% range was comprised exclusively of 16 cases for the prediction of “death or severe BPD”, and a single case for the prediction of “death before discharge”. The increase in calibration error was associated with survival in these rare cases. In all other groups, which encompassed the majority of infants, the fit was found to be good. We considered ECEs of around 3% to be sufficiently small for practical purposes in predicting death or severe BPD among preterm infants immediately after birth.

5. L218, I don't think that the categorical variables are represented in gray.

Answer: 

As the categorical variables did not rank within the top 12 factors, mentioning them would be superfluous. Accordingly, the sentence has been omitted from the manuscript.

6. Fig 2: How did the authors select 20 from 46 variables? I found the data of Figs S4 and S5, but several factors, including neonatal and NICU beds and headcount of neonatologists, seem not related to SHARP value positively or negatively. In contrast, HDP clearly negatively affected SHARP values, but I could not find any comments on the results, even in the discussion.

Answer: 

In each model, we selected the top 12 factors based on their mean absolute SHAP values. For some prognostic factors, no consistent association was observed between the values and their SHAP values. There may be complex interactions that cannot be fully elucidated by the SHAP dependency plots. We have added explanations regarding this to the revised manuscript to improve clarity.

Diverse pathologies, including HDP, congenital anomalies, infections, and placenta previa, are known to increase the risk of preterm birth. The research on preterm infants is inevitably conditioned by the occurrence of preterm birth, which acts as a collider for the risk of prematurity. As in the case known as Berkson’s paradox [5], preterm infants born to mothers with HDP are less likely to have other risk factors for preterm birth, including unmeasured ones. While the difficulty of providing a concise explanation for this led to the initial omission of these points, we have added a detailed elaboration in the revised manuscript.

Page 20, Lines 227-228 (Results)

Fig 2 shows the SHAP summary plots, which reveal the top 12 factors with the highest mean absolute SHAP values in each model, demonstrating their impact on the prediction.

Page 20, Lines 232-234 (Results)

The SHAP dependence plots, as illustrated in S5–7 Figs, are two-dimensional scatter plots created for each of the 12 leading factors identified in the SHAP summary plots.

Page 21, Lines 240-241 (Results)

Although the number of neonatal beds and neonatologists also influenced the prediction, the association with SHAP values was not distinct.

Page 25, Lines 320-324 (Discussion)

Studies of preterm infants intrinsically and inevitably involve conditioning by the occurrence of preterm birth, which acts as a collider for the risk of preterm birth such as non-reassuring fetal status, congenital diseases, infections, and placental abruption. Lower SHAP values in mortality prediction among infants with maternal HDP may be due to the reduced incidence of other preterm delivery risk factors, including unmeasured ones.

Page 25, Lines 331-332 (Discussion)

The inconsistent impact of the number of neonatal beds and neonatologists on the “death or BPD” prediction may suggest the presence of complex interactions with other predictive factors.

7. L223-224, 227-228: Please move the definition of variables from result to methods.

Answer: 

In accordance with your comment, the location of the text has been changed accordingly.

Page 9, Lines 136-140 (Materials and methods)

Respiratory distress syndrome (RDS) was diagnosed based on clinical symptoms, chest X-ray, and microbubble test. Persistent pulmonary hypertension of the newborn (PPHN) was defined as pulmonary hypertension and right-to-left shunt blood flow diagnosed by echocardiography in the acute phase.

8. L231, S5.Fig. How were these variables evaluated as crucial predictors?

Answer: 

We generated SHAP dependence plots (S5–7 Figs) for the 12 prognostic factors in the SHAP summary plot (Fig 2). To improve clarity, an explanation has been added to the revised manuscript.

Page 20, Lines 232-234 (Results)

The SHAP dependence plots, as illustrated in S5–7 Figs, are two-dimensional scatter plots created for each of the 12 leading factors identified in the SHAP summary plots.

9. L243, S7.Fig: HDP seems to be forming a distinct branch. Why was it ignored?

Answer: 

We refrained from mentioning HDP to avoid redundancy as there was no apparent association with the clusters shown in Fig 3. In light of your comment, we have added an explanation to the revised manuscript to improve clarity.

Page 22, Lines 270-271 (Results)

The high-volume facilities in the death or BPD prediction and HDP in mortality prediction also formed their own bunches.

10. In relation to question 6, please discuss the effect of HDP, antenatal corticosteroids, and the facility characteristics on the outcome, too.

Answer: 

In accordance with your comment, we have added the following details to the revised manuscript:

Page 21, Lines 240-243 (Results)

Although the number of neonatal beds and neonatologists influenced the prediction, the association with SHAP values was not distinct. S5 Fig shows that SHAP values consistently decrease with increasing birth weight and gestational weeks; moreover, it also indicates pessimistic predictions for facilities with fewer than 25 annual VLBW infants.

Page 21, Lines 250-251 (Results)

Furthermore, facilities with fewer than 20 neonatal beds were also identified as a risk factor for “death or severe BPD”. 

Page 21, Lines 254-256 (Results)

Antenatal steroid administration and hypertensive disorders of pregnancy (HDP) were associated with lower SHAP values.

Page 22, Lines 259-261 (Results)

The higher the number of full-time nurses in the neonatal ward, the more optimistic the predictions for mortality in extremely preterm infants became.

Page 25, Lines 320-332 (Discussion)

Studies of preterm infants intrinsically and inevitably involve conditioning by the occurrence of preterm birth, which acts as a collider for the risk of preterm birth such as non-reassuring fetal status, congenital diseases, infections, and placental abruption. Lower SHAP values in mortality prediction among infants with maternal HDP may be due to the reduced incidence of other preterm delivery risk factors, including unmeasured ones. The optimistic impact of antenatal steroid administration on mortality prediction may have been enhanced by the implication that the preterm delivery was not entirely unforeseeable. As for facility factors, fewer annual admissions of VLBW infants, neonatal beds, and full-time nurses were risk factors for “death or BPD”, “death or severe BPD”, and “death before discharge”, respectively. Given that infants admitted to high-volume facilities are likely to have a higher severity of illness, these findings may suggest the importance of a skilled and sufficient workforce to improve outcomes. The inconsistent impact of the number of neonatal beds and neonatologists on the “death or BPD” prediction may suggest the presence of complex interactions with other predictive factors.

Thank you for your consideration. I look forward to hearing from you.

Sincerely,

Yoshimasa Kawazoe

Artificial Intelligence and Digital Twin in Healthcare, Graduate School of Medicine, The University of Tokyo, Japan, 

Department of Healthcare Information Management, The University of Tokyo Hospital, Tokyo

7-3-1 Hongo, Bunkyo-ku, Tokyo 113-0033, Japan, 

Tel: +81-3-5800-9077

Fax: +81-3-3813-7238

Email: kawazoe@m.u-tokyo.ac.jp (YK)

---

## [Decision Letter · Decision Letter 1]

6 Mar 2024

Immediate postnatal prediction of death or bronchopulmonary dysplasia among very preterm and very low birth weight infants based on gradient boosting decision trees algorithm: A nationwide database study in Japan

PONE-D-23-39831R1

Dear Dr. Kawazoe,

We’re pleased to inform you that your manuscript has been judged scientifically suitable for publication and will be formally accepted for publication once it meets all outstanding technical requirements.

Kind regards,

Kazumichi Fujioka

Academic Editor

PLOS ONE

Additional Editor Comments (optional):

Reviewers' comments:

Reviewer's Responses to Questions

**Comments to the Author**

1. If the authors have adequately addressed your comments raised in a previous round of review and you feel that this manuscript is now acceptable for publication, you may indicate that here to bypass the “Comments to the Author” section, enter your conflict of interest statement in the “Confidential to Editor” section, and submit your "Accept" recommendation.

Reviewer #1: All comments have been addressed

Reviewer #2: All comments have been addressed

2. Is the manuscript technically sound, and do the data support the conclusions?

Reviewer #1: Yes

Reviewer #2: Yes

3. Has the statistical analysis been performed appropriately and rigorously? 

Reviewer #1: Yes

Reviewer #2: Yes

4. Have the authors made all data underlying the findings in their manuscript fully available?

Reviewer #1: Yes

Reviewer #2: Yes

5. Is the manuscript presented in an intelligible fashion and written in standard English?

Reviewer #1: Yes

Reviewer #2: Yes

6. Review Comments to the Author

Reviewer #1: The authors have addressed the comments made by the reviewers in an adequate matter. Because this paper is based on the NRNJ network data, it is understandable some limitations exist in the in depth analysis of the findings. It would be helpful to add objective measures of PPHN in the network CRF for future studies in the near future.

Reviewer #2: I think that the authors have solved all of my questions. They have successfully shown the results.

7. PLOS authors have the option to publish the peer review history of their article (what does this mean?). If published, this will include your full peer review and any attached files.

Reviewer #1: No

Reviewer #2: No

---

## [Editor Report · Acceptance letter]

14 Mar 2024

PONE-D-23-39831R1 

PLOS ONE

Dear Dr. Kawazoe, 

I'm pleased to inform you that your manuscript has been deemed suitable for publication in PLOS ONE. Congratulations! Your manuscript is now being handed over to our production team.

Kind regards, 

on behalf of

Dr. Kazumichi Fujioka 

Academic Editor

PLOS ONE